Prevalence and type distribution of human papillomavirus in a Chinese urban population between 2014 and 2018: a retrospective study

Xu Mei-Yan 1
Cao Bing 2
Chen Yan 3
Du Juan 4
Yin Jian 1
Liu Lan 721Lancy@sina.com 5
Lu Qing-Bin qingbinlu@bjmu.edu.cn 4 6
1 Department of Nutrition, Aerospace Center Hospital , Beijing , China
2 Key Laboratory of Cognition and Personality (SWU), Faculty of Psychology, Ministry of Education, Southwest University , Chongqing , China
3 Dalla Lana School of Public Health, University of Toronto , Toronto , ON , Canada
4 Department of Laboratorial Science and Technology, School of Public Health, Peking University , Beijing , China
5 Department of Health Management, Aerospace Center Hospital , Beijing , China
6 Beijing Key Laboratory of Toxicological Research and Risk Assessment for Food Safety , Beijing , China
Mossong Joël
Electronic publication date: 2020 Mar 23
Publication date: 2020
Volume: 8
Electronic Location ID: e8709
Received 2019 Sep 9; Accepted 2020 Feb 7
Copyright: ©2020 Xu et al.
Copyright year: 2020
Copyright holder: Xu et al.
License: This is an open access article distributed under the terms of the Creative Commons Attribution License, which permits unrestricted use, distribution, reproduction and adaptation in any medium and for any purpose provided that it is properly attributed. For attribution, the original author(s), title, publication source (PeerJ) and either DOI or URL of the article must be cited.
License URL: https://creativecommons.org/licenses/by/4.0/

Keywords: Human Papillomavirus, Prevalence, Genotype, Screening, Infection, China, Urban

Funding: National Natural Science Foundation of China 81703274 China Mega-Project for Infectious Diseases Grant 2017ZX10103004 Youth Innovation Funding 2014QN09 Peking University Medicine Seed Fund for Interdisciplinary Research BMU2018MX009 The study was supported by National Natural Science Foundation of China (No. 81703274), the China Mega-Project for Infectious Diseases Grant (2017ZX10103004), the Youth Innovation Funding (2014QN09) and the Peking University Medicine Seed Fund for Interdisciplinary Research (BMU2018MX009). The funders had no role in study design, data collection and analysis, decision to publish, or preparation of the manuscript.

==============================
Background

Human papilloma virus (HPV) infection is one of the most common sexually transmitted infections among women worldwide. The current study’s main objective is to report the prevalence and distribution of HPV types in an urban population in Beijing, China.

Methods

All the eligible female participants aged ≥18 years were recruited from the Aerospace Center Hospital in Beijing, China between 2014 and 2018. A total of 21 HPV types were detected by the polymerase chain reaction (PCR) reverse dot blot method and fluorescence quantitative PCR method.

Results

In total, 12 high risk HPV types and nine low risk HPV types were detected. The HPV-positive rates were 8.85% in 2014, 7.16% in 2015, 7.60% in 2016, 8.31% in 2017, and 7.72% in 2018, respectively, in an urban population in Beijing, China. Overall, no significant differences in the HPV-positive rates were found over the five years. The peak prevalence of HPV infection in all types was observed in age group of 20–24 in all types. HPV52 was the dominant HPV type across the five years . Among all 21 HPV types, HPV66, HPV26, and HPV59 were ranked the top three in coinfection occurrence.

Conclusions

Our findings are very helpful for HPV screening and vaccination. The associations between gynaecological diseases and the HPV types with high prevalence, particularly HPV52, warrant further investigation.

Introduction

Human papilloma viruses (HPV) are small groups of double-stranded DNA viruses without enveloped icosahedral capsids (Boda et al., 2018; Zur Hausen, 2002) that cause the most common sexually transmitted infections among women worldwide (Handler et al., 2015; Satterwhite et al., 2013). More than 200 HPV types have been identified (Ghedira et al., 2016) and some had been confirmed to be associated with certain diseases, mainly cancers. Replicated evidence support that HPV infection causes cervical cancer in virtually 100% of cases and it is associated with the development of an important percentage of penile, vaginal, anal, vulvar, and oropharyngeal cancers and pre-cancers (Bansal, Singh & Rai, 2016). Epidemiological research has reported that 5.2% of all cancers worldwide can be attributed to HPV infection (Steben & Duarte-Franco, 2007). Moreover, the overall annual direct medical cost burden of preventing and treating HPV-associated diseases was estimated as an annual $8.0 billion in the United States in 2010 (Chesson et al., 2012).

Each of the HPV type is identified with a number corresponding to the sequence of discovery. The International Agency for Research on Cancer (IARC) working group classified the following12 HPV types as carcinogenic to humans (Group 1) and high risk HPV (HR-HPV): including HPV16, HPV18, HPV31, HPV33, HPV35, HPV39, HPV45, HPV51, HPV52, HPV56, HPV58, and HPV59 (Clifford, Howell-Jones & Franceschi, 2011).

HPV type prevalence varies between different countries, but also within different regions of the same country (Wang et al., 2015). A meta-analysis reported that the estimated global HPV prevalence among women of all ages is 11.7%. The areas with the highest prevalence are Sub-Saharan Africa (24.0%), Eastern Europe (21.4%), and Latin America (16.1%), respectively (Bruni et al., 2010). The prevalence of HPV infections among mainland Chinese women is reported as 11.0% (Zhou et al., 2018), which is similar to the average level worldwide. Additional factors such as age, marital status, and underlying coinfection such as with human immunodeficiency virus (HIV), were reported to be associated with HPV infections (Shi et al., 2016).

HPV prevalence is still a concern since there is no HPV vaccine that covers all HPV types associated with cervical cancer. Previous epidemiological studies have mainly focused on the prevalence of HR-HPV types with specific cancers, updated information on type-specific HPV prevalence and distribution in a general population is warranted. Therefore, we conducted a retrospective, hospital-based study to estimate the overall prevalence and distribution of HPV types in Beijing, China. The persistence rates of HPV infection after a 3-year follow up were also determined. The current findings will be valuable in the development and application of HPV vaccines in China.

Methods

Study design and participants

For this hospital-based retrospective epidemiological study, participants were recruited from the Aerospace Center Hospital located in Beijing, China between 2014 and 2018. The inclusion criteria of our subjects were as follows: (1) female, age 18 years old or older; (2) participating in annual health examinations and having HPV detected result at least once at the current hospital during our research period. We excluded (1) pregnant and postpartum women; and (2) the people who have been diagnosed with any kind of cancer. For our research samples, some had participated in annual health examinations every year, and some only participated once during our study period. The subject had a healthy examination once a year and had one test for HPV infection. We included all available data in the statistical analysis. We conducted a chart review to collect the recruited subjects’ age.

The research protocol was approved by the human ethics committee of the Aerospace Center Hospital and all methods were carried out in accordance with the approved guidelines (No. 2014YN-01). All participants provided written or verbal informed consent to have their samples and information collected and used for this study.

Outcome measurement

Cervicovaginal cellular swabs were collected by a doctor during gynecological healthy examination. Most samples were tested on the same day after collection; otherwise, they were stored at −70 °C until tested.

Viral DNA was isolated from cervicovaginal cellular swabs using a QIAamp DNA Mini Kit (QIAGEN). The 15 HPV types were detected using the polymerase chain reaction (PCR) reverse dot blot method based on L1 region with the HPV Genotyping Kit (Yaneng Biotechnology (Shenzhen) Co., Ltd, Shenzhen, China), which enabled the detection of 12 HR-HPV types (HPV16, 18, 31, 33, 35, 39, 45, 51, 52, 56, 58 and 59) and three low risk HPV (LR-HPV) types (HPV6, 11 and 68) during 2014–2016. An aliquot of a 5 µL DNA sample was briefly used, and the PCR reaction was amplified in a thermal cycler under the following conditions: an initial 15 min at 50 °C, 10 min at 95 °C; 40 cycles of 30 s at 94 °C, 90 s at 42 °C, and 30 s at 72 °C; and a final extension 5 min at 5 °C. The PCR products were immobilized onto a nitrocellulose membrane and hybridized with typing probes. Final results were judged by the direct visualization of the location of blue spots located on the membrane. During 2014-2018, 21 HPV types were detected using the fluorescence quantitative PCR method with a nucleic acid detection kit (Jiangsu Shuoshi Biotechnology Co., Ltd, Taizhou, China) that enables the detection of 12 HR-HPV types (HPV16, 18, 31, 33, 35, 39, 45, 51, 52, 56, 58 and 59) and nine LR-HPV types based on E region (HPV06, 11, 26, 53, 58, 66, 73, 81 and 82; HPV26, 53, 66, 73 and 82 were considered as LR-HPV types in this study). Each PCR reaction was performed briefly in a 40 µL mixture containing 4 µL of extracted DNA and 36 µL of PCR master mix, at 94 °C for 2 min, 40 cycles of 10 s at 93 °C, and 30 s at 62 °C. Specific primers and correspondent fluorescent probes were designed to detect 21 HPV types in the assay. A single copy gene was amplified to serve as an internal quality control for DNA preparation. A sample was determined as HPV positive when the cycle threshold (t value) was less than or equal to 38.0, and the amplification curve was a typical S-type. According to the preliminary experiments in the manuals, the sensitivity and specificity in both the two kits were more than 98% (http://www.yanengbio.com/HPV/ and http://www.s-sbio.com/product/388.html). Detailed information on the two detection methods is listed in the Table S1.

Statistical analysis

Annual HPV prevalence rates and the distribution of HPV types in each year were calculated using positive rates or proportions and 95% confidence intervals (CIs), and they were stratified by age at diagnosis. We explored the differences of HPV detection across age using either an independent t-test. A positive HPV sample was defined as having an infection of HPV types equal or more than one. A single infection was defined as being infected by one type of HPV infection. The coinfection proportion was defined as the number of subjects infected with more than one HPV type divided by the total number of subjects infected with at least one HPV type. The persistence rate of HPV infection was calculated as (①+ and ②+)/(①+ and ②+/-) for one year persistence, (①+ and ②+ and ③+)/(①+ and ②+/- and ③+/-) for two-year persistence and (①+ and ②+ and ③+ and ④+)/(①+ and ②+/- and ③+/- and ④+/-) for three-year persistence, where ①, ②, ③, and ④ mean the detection at the first, second, third and fourth year, respectively, and + means HPV positive, and +/- means HPV positive or negative. A two-sided p value of less than 0.05 was considered significant. All statistical analyses were done using Stata 14.0 (Stata Corp LP, College Station, TX, USA).

Results

Overall prevalence and basic characteristic distribution of HPV infection

Between January 1, 2014 and August 31, 2018, 5,880 women involved in this study had at least one HPV detection (Fig. S1). The annual age distributions of the participants every year are shown in Fig. S2. The age distribution was similar across every year and the majority of participants were between 30–45 years old. As shown in Fig. 1, the HPV positive rates were 8.85% in 2014, 7.16% in 2015, 7.60% in 2016, 8.31% in 2017, and 7.72% in 2018, respectively. No significant differences in HPV positive rates were found over the five years (p = 0.357). The median age of the participants between HPV (+) and HPV (−) cases were comparable in the five years, respectively (Table 1).

HPV type distribution

Twelve HR-HPV and nine LR-HPV types were detected, the HR-HPV rate decreased over time (Fig. 2A). The detection rate of LR-HPV types increased, which may be caused by the dissimilarity of the assays, although the detection rates for HPV06 and HPV11 were comparable between the two HPV detection methods (Table S1). The HPV positive rates varied across different age groups. Here we identified the age distribution of HPV infection. As shown in Fig. 2B, most HPV infections occurred in age groups of 20–24 years old, 65–69 years old, and 50-54 years old. Young women aged 20–24 years had the highest LR-HPV and HR-HPV infection rates (Figs. 2C and 2D), and the detection rates of HPV06 and HPV11 also peaked in the 20–24 years old age group (Fig. 2E).

Figure 1 The prevalence rate of HPV from 2014 to 2018.

Table 1 The basic information of the participants recruited at the different years in the study.

Variable	Total	HPV (+)	HPV (-)	P	
Age, years, median (interquartile range)		
2014	43 (35–56)	46 (36–55)	43 (35–56)	0.380	
2015	50 (38–64)	52 (39–63)	50 (38–64)	0.869	
2016	45 (36–58)	45 (36–56)	45 (36–58)	0.351	
2017	46 (36–60)	45 (35–56)	46 (36–60)	0.092	
2018	44 (36–57)	45 (37–58)	43 (36–57)	0.330	

The distributions of positive rate of all 21 detected HPV types were different in each year (Fig. 3 and Fig. S3). The most dominant HPV type was HPV52, which occupied the top position all five years. HPV58, HPV16, and HPV53 were also ranked high but their order slightly changed each year.

Coinfection and persistence of HPV types infection

Multiple HPV infections were very common, and the ranking of HPV type by coinfection status is shown in Fig. 4. About 75% of the HPV types had a coinfection proportion ≥ 50%. Among all 21 HPV types, HPV66, HPV26, and HPV59 were ranked the highest in coinfections. HPV52 had the most positive rate but was ranked the lowest. The distribution of double HPV coinfections is listed in Table S2.

Figure 2 The prevalence rate of low risk and high risk HPV genotypes in different age groups.

(A) prevalence rate of low risk and high risk HPV genotypes in each year; (B) prevalence of all HPV genotypes of different age groups; (C) prevalence of low risk HPV genotypes of different age groups; (D) prevalence of high risk HPV genotypes of different age groups.

Figure 3 The distributions of every HPV genotypes in each year from 2014 to 2018.

Figure 4 Multiple HPV infections and the ranking of coinfection status of each HPV genotype.

Table 2 shows the HPV positive rates in the follow-up periods after the first HPV positive test. Significant differences in HPV positive rates were found in both the persistent infection group and the non-persistent infection group during the 3-years follow-up (p < 0.001). Persistent infections of different HPV types in four different years were detected in 16 cases. As shown in Fig. S4), HPV52, HPV58, HPV35, and HPV68 were more likely to cause persistent infections.

Discussion

The main objective of this retrospective study was to report the prevalence and distribution of HPV types based on the annual health examinations and HPV detection results of women aged ≥ 18 years between 2014 and 2018 at the Aerospace Center Hospital in Beijing, China. The second objective of the study was to with more information in order to protect women from further HPV infections and related cancers in China. Most previous studies mainly focused on HR-HPV types (Guardado-Estrada et al., 2014; Nielsen et al., 2009). To the best of our knowledge, this is the first large-scaled study to provide the prevalence, distribution, and persistence of both HR-HPV and LR-HPV types in an urban population in China.

The prevalence rate of HPV infection revealed no changes over the years, which means that HPV infection exists steadily and persistently, sustainably damaging women’s health. The prevalence rate of HPV infection can be controlled and reduced through the use of HPV vaccines, making the promotion and acceptance of HPV vaccines crucial.

Table 2 The HPV positive rates in the follow-up for different periods from the first HPV positive.

Persistent period	Positive number	Total number	Positive rate (95% CI)	P	
One year	186	218	85.3 (79.9–89.7)	<0.001	
Two years	73	118	61.9 (52.5–70.6)		
Three years	73	118	61.9 (52.5–70.6)		
Notes.

The persistence rate of HPV infection was calculated as (①+ and ②+)/(①+ and ②+/-) for one year persistence, (①+ and ②+ and ③+)/(①+ and ②+/- and ③+/-) for two-year persistence and (①+ and ②+ and ③+ and ④+)/(①+ and ②+/- and ③+/- and ③ +/-) for three-year persistence, where ①, ②, ③, and ④ mean the detection at the first, second, third and fourth year, respectively, and + means HPV positive, and +/- means HPV positive or negative.

The prevalence rate of HR-HPV types shows a slightly decreasing trend overall in all ages over the five years between 2014 and 2018. The detection rate of LR-HPV types, however, shows an overall increasing trend, which may be expected due to the dissimilarity of the assays. Therefore, the variety of HPV infections were increased, although the prevalence rate of HPV infection was reduced. In regard to the distribution of HPV infections across different age groups, the peak prevalence of all HR-HPV and LR-HPV types of HPV infections was observed in the 20–24 years old age group. The prevalence of LR-HPV types in women aged 20–24 years was close to 6%, which was significantly high compared to less than 2% observed in other age groups. Women aged 65–69 years also showed significantly high prevalence in all HR-HPV types. However, these results may be biased by the different detection methods used in this study.

Because of the large population base of China, there is imbalanced development and inconsistent allocations of healthcare resources and public health awareness (Qiao, 2018). A previous study showed that women aged 40-49 years were the peak age group for HPV infection prevalence, accounting for 17.9% of cases in in Ningbo, south-eastern China (Baloch et al., 2017). According to a study performed in Yunnan, south-western China, women with the highest HPV prevalence were ≤ 29 years old in the urban regions and ≥ 50 years in rural regions. These studies did not achieve uniform results on age distribution (Baloch et al., 2017). In addition to age, location (i.e., rural/urban areas) and healthcare resources, other risk factors may contribute to the prevalence and distribution of HPV, such as marital status, sexual activity, genetic variants, and coinfections with other diseases (i.e., HIV) (Zhao et al., 2018), and education, public health awareness, alcohol use, and tobacco use may be potential confounding variables (Baloch et al., 2017).

Over the five-year period, HPV52 was the most dominant type among patients with positive detections, supporting the findings of several other studies (Baloch et al., 2017; Hong et al., 2015). It is followed by HPV58 and HPV16 as the most common types in female residents in Shanghai (Xu et al., 2018). In addition to single infections caused by HPV types, HPV52 also accounted for over 80% of coinfection cases, followed by HPV58 and HPV16. HPV26, HPV73 and HPV82 had the lowest single infection/coinfection proportions. The high prevalence of HPV52 in our study population and its association with other gynecological diseases warrant further investigation.

According to the previously reported data, over 70% of women have had an HPV infection at least once in their lifetime, with about 10% having a lifelong persistent infection (Liao et al., 2018). Persistent HPV infection can cause abnormal proliferation of cells, and the accumulations of genetic damage leading to cancer of the cervix, vulva, vagina, anus, penis, and/or oropharynx (Crosbie et al., 2013; Hong et al., 2015). It is the second most common cause of death from cancer for women globally (Lowy & Schiller, 2006), with higher death rates reported in lower income countries (Qiao, 2018). According to our findings, persistent infection contributed to the positive rate of HPV detection over a three-year follow-up. High-risk HPV16 and HPV18 account for over 70% of cases of cervical cancer, which is the fourth most common cancer among women worldwide (Qiao, 2018), while LR-HPV06 and LR-HPV11 account for about 90% of external genital warts cases (Lowy & Schiller, 2006).

Prevention methods such as screening, early-detection and treatment, and vaccination are used to fight HPV viruses (Qiao, 2018). Three types of vaccinations are available in mainland China: the bivalent vaccine which targets HPV16 and HPV18; the tetravalent vaccine which targets HPV6, HPV11, HPV16 and HPV18; and the nine-valent (9v) vaccine targets five additional HR-HPV types (HPV31, HPV33, HPV45, HPV52 and HPV58) (Crosbie et al., 2013; Lowy & Schiller, 2006). The 9v vaccine (Gardasil 9) was approved by the Food and Drug Administration of China and was recently released into the market on April. 28, 2018. A set of three vaccines costs about 5,800 CNY, which is about twice the price in North America countries (i.e., America, Canada). Due to limited quantities imported from the United States to a high number of clinics and hospitals, vaccines are scare in China. Women 25 years and older may be able to afford the cost and long wait time, but younger women may not be able to afford the vaccines themselves if their healthcare programs do not cover the cost. Implementing HPV prevention initiatives may not meet the current need (Ogembo et al., 2015).

All data in the current study are based on highly reliable hospital records. The large sample size-based population means the results of this study are generalizable. Limited data are available on the prevalence and type distribution of HPV infections in Beijing, the capital of China making the present study valuable for the future clinical researches of HPV screening and anti-HPV vaccines. HPV distribution largely varies and is affected by many confounding variables. The results and interpretations presented in the current study should be considered in the context of the following limitations. First, all participants were recruited from the Aerospace Center Hospital in Beijing and most of them were from urban areas, which is not sufficiently representative of the whole population and may be selection biased. Second, since individuals in China can choose their type of physical examination from a variety of packages and can voluntarily choose or decline HPV type testing, a large number of people do not have the results from all five years. Additionally, we can only perform analysis of persistent infection in a small number of participants with follow-up test results. A detection bias for some HPV types exists in the study due to changing and unbalanced distributions of detection methods, especially for the LR-HPV types, which may be related to changes in LR- or HR-HPV types.

Conclusion

The current retrospective study presents the prevalence and distribution of HPV types in women between 2014 and 2018 in an urban population of Beijing, China. HPV52 was the most dominant HPV type over all five years. The peak prevalence of all HPV infection types was observed in ages 20–24 years old in all types. The current findings can be significant for the future of HPV screening and vaccination. The associations between gynaecological diseases and HPV types with high prevalence, especially HPV52, warrant further investigation.

Supplemental Information

Figure S1 The diagram flowchart of the participants

Click here for additional data file.

Figure S2 The prevalence rate of low risk and high risk HPV types in different age groups

(A) prevalence rate of low risk and high risk HPV types in each year; (B) prevalence of all HPV types of different age groups; (C) prevalence of low risk HPV types of different age groups; (D) prevalence of high risk HPV types of different age groups; (E) prevalence of HPV06 and HPV11.

Click here for additional data file.

Figure S3 The distributions of every HPV types in each year from 2014 to 2018

Click here for additional data file.

Figure S4 The persist infections of different HPV types in different four years within 16 participants

Click here for additional data file.

Table S1 Two detection methods in the different years

Click here for additional data file.

Table S2 The distribution of two HPV types coinfection

Click here for additional data file.

Table S3 Raw data

Click here for additional data file.

Table S4 Codebook

Click here for additional data file.

Additional Information and Declarations

Competing Interests

Author Contributions

Human Ethics

Data Availability

The authors declare there are no competing interests.

Mei-Yan Xu and Qing-Bin Lu conceived and designed the experiments, analyzed the data, prepared figures and/or tables, authored or reviewed drafts of the paper, and approved the final draft.

Bing Cao analyzed the data, prepared figures and/or tables, authored or reviewed drafts of the paper, and approved the final draft.

Yan Chen performed the experiments, prepared figures and/or tables, authored or reviewed drafts of the paper, and approved the final draft.

Juan Du performed the experiments, prepared figures and/or tables, and approved the final draft.

Jian Yin performed the experiments, authored or reviewed drafts of the paper, and approved the final draft.

Lan Liu conceived and designed the experiments, performed the experiments, authored or reviewed drafts of the paper, and approved the final draft.

The following information was supplied relating to ethical approvals (i.e., approving body and any reference numbers):

The research protocol was approved by the human ethics committee of the Aerospace Center Hospital and all methods were carried out in accordance with the approved guidelines (No. 2014YN-01). All participants provided written or verbal informed consent to have their samples and information collected and used for this study.

The following information was supplied regarding data availability:

The raw measurements are available in Table S3.

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
