# Peer review of "Prevalence and type distribution of human papillomavirus in a Chinese urban population between 2014 and 2018: a retrospective study"

_PeerJ, doi:10.7717/peerj.8709_

## Round 0.1 · original submission · Major Revisions

· Academic Editor

Major Revisions

All three reviewers agree that your manuscript needs major revisions including editing by a native English writer. Please make sure you address the major issue if the study was based on serum samples or cervical swabs and discuss this thouroughly including comparisons to other studies which have used serum for HPV detection. Further make sure that you describe in more detail how samples were taken and laboratory methods.

Reviewer 1 ·

Basic reporting

Please see my comment in "General comments for the author".

Experimental design

Please see my comment in "General comments for the author".

Validity of the findings

Please see my comment in "General comments for the author".

Additional comments

Dr. Lu and the research group investigated HPV infection over 5 years from a population of women living in an urban area of Beijing, China. A large sample size was assessed (>5000), with results indicating high prevalence of certain high risk HPV types (such as HPV52, 58, 16) in the surveyed area. As an epidemiological study, however, the study design, data presentation and analyses need to be improved before this manuscript could be accepted for publication.

1. The authors should have a diagram showing detailed subjective recruitment, such as criteria of inclusion and exclusion, total women recruited by year, follow-up size and the status of follow-up by year (e.g., drop out, re-visit...), available demographic information.

2. The authors should clearly indicate in abstract or title that serum but not cervicovaginal cellular swabs were used for HPV screening. To me, it is very important since HPV prevalence or type distribution is dramatically different between variable sample types. Meantime, discussion should be addressed for the difference in the manuscript since the authors compared prevalence and type distribution with other studies.

3. As shown in supplemental table 1 and the part of "outcome measurement", the authors used two different kits for HPV DNA screening. I realized, for the latest 3 years (2016-2018), the majority of samples were tested with "fluorescence quantitative PCR" that covers 9 LR-HPV types while "PCR reverse dot blot" used for the first 3 years includes 3 LR types only. It concludes me that the increasing rate of LR HPV infection by year (Figure 2A) was probably not true due to in part the bias of detection methods for LR HPV types. Alternatively, the authors may focus on HPV6/11 alone to compare LR-HPV prevalence.

4. HPV is a kind of sexual transmitted infection which is closely related to life style, sex history, and marriage status. I don't have enough knowledge on the relationship between HPV infection and diabetes and/or hypertension, so the authors are recommended to justify the reason for such kind of statistical test, while the lack of key demographic information could largely impair the significance of this work.

5. I don't how to read Table 1. For example, Age (2014) - 43 is mean age? if so, what's meaning of 105 and 9.1 of the first part of diabets for example. How to define positive and negative status of diabetes and hypertension, and what's the proportion among the surveyed population?

6. Table 2, do authors have separated data between high-risk and low-risk HPV infection?

7. For Figure 3, the reviewer suggests to combine 2014-2018 with bar columns grouped by HPV types.

8. Please use "type" to replace "subtype" through the whole manuscript.

9. Line 40 - "All the HPV subtypes were detected". Please do not over-estimate HPV detection since only 12 HR and 10 LR HPV types were covered by the screening kits used in this study.

10. The English writing should be improved.

11. I feel confused about Supplemental tables 3 and 4.

Reviewer 2 ·

Basic reporting

The manuscript describes the prevalence and type-specific distribution of HPV infection in 5880 women over a 5-year interval. Authors also investigate persistence rate and coinfection in the population. It is quite interesting to see type-specific HPV distribution in this population, especially dominance of HPV52, while HPV18 is not even in the top five. Although the present manuscript has some major shortcomings, it would contribute to the current literature and thereby should be published.
1. First, manuscript requires editing by professional/native English medical writer. Apart from grammatical mistakes throughout the manuscript, it is not always clear what authors are trying to say. In addition, I have some major concerns regarding the methodological approach and interpretation of the results/conclusions.
2. The methodology of the HPV detection kits used is poorly described, could you please provide more details. I could not find information by the online reference provided. Furthermore, one of the links provided is accessible only in the Chinese language.
3. According to the Supplemental Table 1, first 3 years mainly PCR reverse dot blot method was used, which was stopped in 2017 and replaced by fluorescence quantitative PCR. The first method detects 12 high-risk and three low-risk HPV types, while the second method detects 12 high-risk and 9 low-risk. It seems to me that the authors did not take into account this difference when interpreting the results. Significant increase in low-risk HPV detection appears when fluorescence quantitative PCR test was used frequently. You would expect this increase given that the first method is limited to detection of 15 HPV types, while the second could detect 21. It is necessary to adjust all analyses and results according to the detection abilities of the methods. To identify the change in HPV prevalence you would need to limit the HPV detection to genotypes detectable by both methods. Similarly, decrease in HPV high-risk prevalence might be related to changes in HPV detection kits, even though both methods are able to detect 12 high-risk genotypes. I would like to draw the attention to the authors to tackle this in a detailed manner to limit the spurious findings that might arise with the methodology presented. In addition, these major limitations are not mentioned in the discussion.
4. In regards to statistical analysis, the association between demographic and clinical variables cannot be studied with t-test or Chi2. These test just show if the groups are statistically different from each other, therefore more complex analysis would be necessary to study an association. Although, it is not clear to me how diabetes or hypertension might affect HPV infection incidence.
5. HPV persistence and clearance is not well defined as well as how coinfection proportion. Could you please make it clearer?
6. Could you please calculate median clearance and median persistence per type? Results presented in Table 2 are a bit confusing. In case you do not have enough data to show clearance/persistence by type, I would suggest grouping, for example according to phylogenetic distance (e.g. alpha 7, alpha 9).

Minor issues:
Abstract: Line 40 authors stated, “all the HPV subtypes were detected”. What do you mean by that?
Does “subtype” mean HPV type, or could you detect subtype/lineage of particular HPV type?
Please, use word type or genotype when referring to the HPV types.

Introduction:
Line 60-61: There is no evidence of an increased risk of cancer when infected with multiple HPV genotypes.
Line 68-71: 12 high-risk genotypes are mentioned, but only 13 are listed. HPV66 is classified as possibly carcinogenic by IARC.
Methods:
Line 97: Did you also exclude women with cervical cancer? If that is the case, how many women were excluded? Since this is HPV prevalence study, I would suggest keeping women with cervical cancer cases. Did participants receive a pap test? It would be interesting to see the impact of HPV incidence/persistence on pap-test results.
Line 123: Was any other clinical or demographic information collected?
What is the reason for women coming to the hospital? Do they have a health issue at admission or it is just a check-up? Were women invited for the examinations?
Do you have any information on HPV vaccination of young women participating in this study?
Why did you switch to fluorescence quantitative PCR test?
How was hypertension defined?
What was the median follow-up time?
How many participants received more than one examination?
Results: Line148-154, Please see comment 3 above. Did you take into account multiple testing when reporting HPV prevalence? Re-testing HPV positive women would have an impact on reported incidence.
Line 155-160: Again, see comment 3 above. The first kit can detect 15 genotypes, while the second can detect 21. You would need to limit these results to 15 genotypes detectable by both methods, or report separate results for each kit used.
How was the coinfection proportion calculated? Please define in methods.
What was HPV acquisition rate? The proportion of participants negative at baseline and positive at follow-up?
Line 167-172: It is a bit difficult to understand HPV persistence results. May be reporting median clearance and persistence time in months would make it easier for the reader.
Line 169:
Discussion.
Line 181: Confounders cannot be studied with the statistical tests described in methods, see comment 4.
Line 184-193: See comment 3 above.
Line 241: I have doubts regarding the generalizability of the results of the current study. It is not clear to me why these women are attending the hospital, are they invited, or they attend due to previous health issues? If they are invited, what is the participation rate? Moreover, as you mentioned above in the discussion, HPV distribution is largely affected by many confounders and varies across the areas.
Line 253: You need to address the issue of changes of the detection methods in more details and how this would impact the results.
Since the first method was not used from 2017, I assume there are some participants tested with the first method and re-tested with the second. Could you please discuss this fact? Would this affect HPV persistence results?

Experimental design

See above

Validity of the findings

See above

·

Basic reporting

- The authors described the prevalence of HPV DNA and their subtypes in a population of China with very robust number of specimens. It is important to mention that HPV prevalence still a concerning since there is no HPV vaccine which covers all HPVs associated with cervical cancer.
- I really recommend updating the introduction with respect to IARC classification. It is important to mention that some HPVs are also classified as possible or probably HR, which still are necessary evidences of their oncogenesis (based on molecular or epidemiological evidences). The author says 12 HR-HPVs but 13 was written, please, re-check the literature (PMID: 30963559 and others)
- There is no changing in HPV DNA prevalence during the years, and this should be better addressed in the discussion.
- The English grammar is good.

Experimental design

- Regarding the study design and participants, I strongly recommend rewriting it. Please, explain better the number of participants included, and the samples. How was collected and stored?
- The authors say that Viral DNA was isolated from serum samples, however, only few studies have reported presence of HPV DNA into the bloodstream of patients. This must be justified and clarified and very well discussed since could strongly impact the results. Due to this fact, I recommend changing the Title, including the type of specimen used.
- Should be clarified in the text and compared the HPV prevalence with studies that used the same technique and the limitations of this study.

Validity of the findings

The descriptive results are well addressed.

---

## Round 0.2 · Major Revisions

· Academic Editor

Major Revisions

The English of the manuscript remains unacceptable. Please have the manuscript completely revised by native English speaker and scientific writer. Only then will the manuscript be sent to the reviewers for comments.

---

## Round 0.3 · Major Revisions

· Academic Editor

Major Revisions

Please address all remaining issues identified by reviewer 2, in particular on the low risk genotypes.

Reviewer 1 ·

Basic reporting

I have no further comment.

Experimental design

I have no further comment.

Validity of the findings

I have no further comment.

Additional comments

I have no further comment.

Reviewer 2 ·

Basic reporting

The manuscript has significantly improved, although I still have some concerns which have not been addressed by the authors.
As I mentioned in my previous review the prevalence of low risk genotypes has increased due to the change of the genotyping assays, it is not an increase in prevalence but simply increase in detection due to the broader spectrum assay used. Therefore, authors should carefully revise the interpretation of their results and discussion.
For example line 169. The statement regarding the significant difference of lr-hpv positivity rate between 2014 and 2018 is misleading and should be removed. This difference is expected due to the dissimilarity of the assays. The comparison would make sense if authors show the prevalence of low risk genotypes covered by both methods such as HPV6 and 11.
Similarly, lines 180-182 authors state that HPV positivity was zero in 2014 for some low risk HPV and increased over time. One would expect such results given that the reverse dot blot method can detect only HPV6 and 11. Please revise or remove the sentence.
Line 210: I would suggest explaining in details that study results showed increased detection rate of lr-hpv rather than increased prevalence, since the fluorescence quantitative PCR targets 9 lr-HPV types compared to 3 lr-HPV with reverse dot blot method.
The discussion would benefit if authors justify why they investigate an association between diabetes/hypertension with HPV infection and interpret results.
Minor changes:
Could authors please specify target region/gene of the assays.
Please briefly mention in abstract detection methods used.
Please address minor grammatical errors throughout the manuscript e.g: line 71, line 91, line 162.
Line 112 and 121 please indicate years. Authors should clearly explain in methods which test was used during which period.
It is still confusing for me how was the persistence rate calculated. Could authors please revise the paragraph.
Some sentences still remain unclear especially in the discussion part e.g. lines 245-247.
Tables and figures:
In table one please indicate in a footnote that the numbers are median and frequencies as in reply to reviewer one comment 5.

Experimental design

see above

Validity of the findings

see above

Additional comments

see above

·

Basic reporting

The authors improved the paper with minor revision regarding english grammar.

Experimental design

The authors compared risk factors of diabetes and hypertension with hpv presence. Would be great if the authors could explain why the compared with these variables, and what the results means. It can be explained in the discussion or results.

Validity of the findings

The results are important in order to keep the screen of hpv in a population where lack these informations.

---

## Round 0.4 · accepted · Accept

· Academic Editor

Accept

Thank you for your revision

Reviewer 2 ·

Basic reporting

The manuscript has been improved significantly.

Experimental design

.

Validity of the findings

.

Additional comments

.